# The Prevalence of Compassion Fatigue among Oncology Healthcare Professionals in Three Public Healthcare Facilities in Kwazulu-Natal, South Africa

**DOI:** 10.3390/ijerph20075412

**Published:** 2023-04-05

**Authors:** Phindile C. Mlaba, Themba G. Ginindza, Khumbulani W. Hlongwana

**Affiliations:** 1Discipline of Public Health Medicine, School of Nursing and Public Health, College of Health Sciences, University of KwaZulu-Natal, Durban 4001, South Africa; 2Cancer & Infectious Diseases Epidemiology Research Unit (CIDERU), College of Health Sciences, University of KwaZulu-Natal, Durban 4001, South Africa

**Keywords:** compassion fatigue, oncology healthcare professionals, cancer, healthcare facilities, KwaZulu-Natal

## Abstract

Compassion fatigue (CF) is a serious global challenge among healthcare professionals dealing with diseases with poor health outcomes in clinical settings. Chronic exposure to the suffering of others is inevitable in the oncology setting and remains one of the main contributors to CF. Therefore, this study determined the prevalence of CF among oncology healthcare professionals (OHPs) in three public healthcare facilities in KwaZulu-Natal, South Africa. This cross-sectional descriptive study was conducted among 73 OHPs using the Professional Quality of Life Scale version 5 questionnaire, and the data were analysed using the Statistical Package for Social Sciences. More than half (56.2%) of the participants reported average scores for CF, with 43.8% of them scoring low. The participants from Inkosi Albert Luthuli Central Hospital had the highest CF mean score (26.8) compared to those from Addington Hospital (21.2) and Greys Hospital (22.9). Female OHPs had a higher mean score (24.3) for CF, compared to their male counterparts (20.6). The CF scores were positively correlated with older age and longer work experience of the OHPs. The prevalence of CF among OHPs was average, compared to those reported by other local and international studies. Nevertheless, these results cannot be taken lightly, given the straining effects of unmanaged CF on the healthcare system generally and on patient care in particular. The results of this study can potentially contribute to policy development and the planning of intervention strategies towards the effective management of CF among OHPs.

## 1. Introduction

Cancer is one of the leading causes of premature mortality among adults and has become a major public health concern, worldwide [1]. This makes oncology one of the fastest-growing medical specialties with a high demand [2]. Continuous exposure to the high cancer morbidity and mortality can be a strenuous experience for oncology healthcare professionals (OHPs), thereby putting them at risk of enduring compassion fatigue (CF) and the associated effects of providing care [3]. CF is defined as a secondary traumatic response resulting from the close contact with the pain and suffering of others [4]. CF is also widely known as secondary traumatic stress or vicarious trauma, and these terms are commonly used interchangeably as they all describe a state of being resulting from the exposure to the trauma of others [5,6]. However, the literature has labelled CF as a more general and user-friendly term [5,7]. In oncology, CF is often experienced through the encounter with recurring patient deaths and the unavoidable exposure to the suffering of patients as well as their grieving families [3,8]. This then takes a toll on the psychological well-being of OHPs and is generally exacerbated by high energy inputs into patient care over extended periods with undesirable health outcomes [9].

CF is progressively becoming a major challenge in the medical field, more so in the field of oncology [8]. The oncology work environment can generate a significant amount of work-related stress, resulting in physical and mental exhaustion, which would inadvertently contribute to employee dissatisfaction [10]. Oncology nurses are at an increased risk of work-related stress and CF, primarily due to the empathy that is felt when patients in their care demise, as they tend to feel a personal sense of failure [10]. This takes a toll on both their professional and personal lives and goes on to affect productivity in the work place [10,11]. This attests to CF having a direct negative effect on the well-being of oncology nurses and consequently that of the patients under their care [12]. Research evidence suggests that oncologists also experience many occupational stressors that are rooted in their profession as cancer healthcare providers [2]. The literature has further revealed that most of the work-related distress among oncologists comes from witnessing patient suffering and dealing with distraught family members of patients who are dying [2,13,14]. These oncology occupational stressors leave oncologists susceptible to CF, as this is a profession that is burdened with frequent patient loss and is likely to affect the emotional well-being of oncologists [15].

Exposure to CF and associated conditions is not limited only to oncology nurses and doctors but involves other allied healthcare professionals, such as radiotherapists, psychologists and social workers, among others [16,17]. These healthcare professionals are also involved in the daily care of cancer patients and thus are OHPs [16,17]. It is against this backdrop that this study determined the prevalence of CF among oncology healthcare professionals in the three public healthcare facilities that offer oncology services in KwaZulu-Natal, South Africa. This study further highlighted the differences in the risk levels of CF among OHPs in relation to the three healthcare facilities, gender, age and work experience in the field. This is to inform the development of more targeted interventional strategies dedicated to the management of CF among OHPs. Given the limited literature on the phenomenon of CF in the South African oncology setting, this study provides a contextual contribution to the broad body of knowledge that exists around the topic of CF.

## 2. Materials and Methods

### 2.1. Study Design

This study was conducted using a descriptive cross-sectional study design to determine the prevalence of CF among OHPs in the three public healthcare facilities that offer oncology services in KwaZulu-Natal (KZN), South Africa [18].

### 2.2. Study Setting

The study was conducted in Addington Hospital (ADH) and Inkosi Albert Luthuli Central Hospital (IALCH) which are located in the city of Durban, and in Greys Hospital (GH) which is in the city of Pietermaritzburg. These are the three public healthcare facilities that offer oncology services in the province of KZN (Figure 1). 

#### 2.2.1. ADH

ADH is a district and regional public hospital located in the city of Durban and is operated by the KZN provincial department of health (DoH) [19]. ADH has a capacity of over 500 beds and approximately 2200 staff members and provides specialised and non-specialised services to patients in the metropolitan area of eThekwini in KZN [19]. The eThekwini metropolitan area is the third largest municipality in South Africa with a population of 3.9 million [20]. eThekwini is labelled as the economic powerhouse of the KZN province with a diversified economy, and over 2 million people live below the upper-bound poverty line [20]. ADH is one of the only three public healthcare facilities that provide oncology services in the province of KZN and one of the only two in Durban.

#### 2.2.2. IALCH

IALCH is a central and tertiary hospital that operates on a referral basis and is also located in the city of Durban in the province of KZN [21]. It is a public and private partnership hospital under the KZN DoH and has a capacity of over 840 beds, with approximately 2500 staff members [21]. IALCH provides a variety of specialist services, including oncology, and is one of the only two public hospitals that provide oncology services in Durban [22]. Although IALCH services patients in the eThekwini Metropolitan area as well, its reach extends to other areas in the KZN province and beyond, as it is a central, tertiary care and referral hospital [21].

#### 2.2.3. GH

GH is a referral and tertiary hospital operating under the KZN DoH and is located in the city of Pietermaritzburg (PMB), which is also the capital city of the KZN province [22,23]. GH has a capacity of 530 beds, with approximately 2100 members of the staff [24,25]. It is the only public hospital that provides oncology services in the city of PMB and is one of three hospitals in the province of KZN [24]. GH services patients from the uMgungundlovu district municipality, which has a population of over 1 million, with an estimated 63.4% of this population living below the poverty line [23,24]. GH offers tertiary services to the western half of KZN, which includes five districts that total a population of 3.5 million [24].

### 2.3. Study Population and Sampling

The study population consisted of all healthcare providers who provided care to cancer patients in the three public healthcare facilities. The healthcare providers are collectively referred to as OHPs and consisted of oncologists, oncology nurses, radiographers, radiation therapist, psychologists, medical officers and other healthcare professionals who regularly worked with cancer patients. Among these were dieticians, physiotherapists, speech therapists, registrars and medical physicist. To be included in the study, OHPs had to have been providing services to cancer patients for at least one year in any of the three participating public healthcare facilities. In consultation with the department management office in the three healthcare facilities regarding the oncology staff composition, approximately 120 OHPs met the eligibility criteria across the three healthcare facilities. Total population sampling was used, whereby the whole population that met the inclusion criteria was included in the study due to the small sample pool [26].

### 2.4. Data Collection Procedure

After ethical approval was sought from the relevant ethics committees (BREC/00002515/2021) (KZ_202103_028), permission to conduct the study was obtained through the management of each healthcare facility. The management office of each healthcare facility was approached in writing, sharing a study summary and all ethical approvals. The three healthcare facilities provided written site permissions, and the research team was introduced to the relevant heads of the oncology departments in the healthcare facilities for access to the potential participants.

The lead researcher was invited to the pre-shift meetings in the oncology departments where the potential participants were informed verbally about the study. This was followed by a distribution of the questionnaire packs which included a study information document containing detailed information about the research study, a consent form and a questionnaire. The completed questionnaires where then collected at the following pre-shift meetings from each participant. Subsequently, other completed questionnaires were collected by the lead researcher as and when they were ready upon communication with the heads of the departments in the three healthcare facilities.

### 2.5. Data Collection Tool

The collection of the data was conducted using a self-administered questionnaire which was administered in the English language. The first and second sections of the questionnaire focused on the demographic data and the Professional Quality of Life Scale version 5 (ProQOL-V), respectively [27]. The ProQOL-V is a 30-item questionnaire that measures CF, burnout and compassion satisfaction [27]. The three sub-scales are each addressed by 10 items in the questionnaire in Likert scale format, from “never” which is represented by the number 1 (one), to “very often” which is represented by the number 5 (five) [27]. In relation to the sample of this study, the Cronbach’s Alpha reliability for the CF subscale was 0.830. The ProQOL-V has well-established reliability and validity and, more especially, construct validity as it is a standard tool that has been used in a substantial number of published articles [3,27].

### 2.6. Data Analysis

The data were analysed through descriptive and inferential statistics, using the Statistical Package for Social Sciences version 28 (SPSS). As guided by the concise ProQOL manual, CF scores were generated by a calculation of the sum of the scores of the 5-point Likert scale responses of the 10 items that measured CF [27].

The total cut-off scores suggested by the concise ProQOL manual for low, average and high CF were:Low CF: ≤22Average CF: between 23 and 41High CF: ≥42

These cut offs were used to determine the prevalence of CF in the three healthcare facilities [27].

Mean scores and standard deviation (SD) for CF were calculated for the categories of gender and healthcare facility. An independent samples t-test was conducted to compare the mean scores for CF based on the category of gender. To compare the mean CF scores between the three healthcare facilities, one-way analysis of variance (ANOVA) was used. For the numeric data of age and work experience, a Pearson correlation analysis was conducted. A *p*-value <0.05 was considered statistically significant.

### 2.7. Ethical Considerations

Approval to conduct this study was obtained from the University of KwaZulu-Natal’s Biomedical Research Ethics Committee (BREC) (BREC/00002515/2021) and the KZN Provincial DoH (KZ_202103_028), respectively. Each of the three healthcare facilities provided support letters granting site permission. All participants signed informed consent forms prior to their participation, after the study had been fully explained verbally and in writing, including the voluntariness and confidential nature of their participation.

## 3. Results

A total of 120 questionnaires were distributed between the three healthcare facilities, with a response rate of 61%. As seen in Table 1, there was a total of 73 participants across the three healthcare facilities. The participants’ age ranged from 23 to 59 years (mean age of 39.16), with 41.1% of the participants within the age category of 31–40 years, and ages for seven participants were missing. The majority (80.8%) of the participants were female. The participants’ occupation varied, with nurses having the greatest number of participants (46.6%). The participants’ years of experience in the field ranged from 1 year to 38 years (mean experience of 11.45 years), and 8 participants had missing data for this category. Among the three healthcare facilities, the majority (46.6%) of the participants were from GH.

Table 2 shows that 32 (43.8%) and 41 (56.2%) OHPs scored low and average for CF, respectively. No participant scored high for CF. There was a significant statistical difference in the CF mean scores between the three healthcare facilities, with *p* = 0.025. The OHPs from IALCH had the highest mean score (26.76) for CF, followed by those from GH (22.85), with those from ADH (mean score = 21.17) scoring the lowest. A post hoc analysis was performed to determine the observed significant difference between the healthcare facilities. The post hoc Tukey test revealed that a significant difference in CF mean scores was observed only between ADH and IALCH, with *p* = 0.027. The mean CF score of female participants (24.25) was higher than that of male participants (20.64), although this difference was not statistically significant (*p* = 0.075) as seen in Table 2.

Table 3 presents the Pearson correlation of the CF scores with age and work experience in oncology. There was a positive correlation of the CF scores with both age and experience, showing that as age and experience increased, so did the CF scores. There was a statistically significant positive correlation between the CF scores and age (*p* = 0.023). However, the positive correlation between the CF scores and experience was not statistically significant (*p* = 0.313).

## 4. Discussion

The overall prevalence of CF among OHPs in the three healthcare facilities ranged from low to average, with no participant scoring high for CF. These findings on CF among healthcare professionals are incongruent with those of other studies from the United States and Greece, which found average to high risk levels of CF [28,29]. Similarly, a study by Mason and Nel conducted in the South African context produced contrasting findings, revealing that a higher proportion of student healthcare professionals had a high risk for CF [30]. This contrast in the results can be attributable to the difference in the study populations, as that study was conducted among nursing students who were still in training in the Gauteng province, while this study was among OHPs who were already established in the healthcare workforce in the KZN province [30]. This study was also conducted during the Corona virus disease 2019 (COVID-19) pandemic, which may have also had an influence on the results. As a response to the COVID-19 pandemic, a national lockdown was implemented, which restricted movement in the country [31]. This saw healthcare facilities reducing services to bare necessity, and oncology services, specifically outpatient follow-up services, experiencing severe curtailment [32,33]. With people being restricted to their homes, the fear of being infected with COVID-19 in healthcare facilities affected health seeking behaviours, resulting in delayed hospital visits [34,35]. This may have resulted in reduced volumes of patients particularly in departments with specialised services such as oncology, further contributing to reduced occupational stress and reduced exposure to the suffering of patients as these are the main contributors to CF [2,4,35]. Therefore, the above factors may provide a plausible understanding of the contrasting results with other CF studies conducted before the COVID-19 pandemic [28,29,30]. The findings of this study are however congruent with those of one other study on CF that was previously conducted among OHPs in the South African context and found CF to be at average levels [3].

The highest CF mean score for IALCH suggests that OHPs from this facility had higher scores for CF as compared to OHPs from ADH and GH. This could be in relation to IALCH being a central, referral and tertiary care hospital and servicing a higher volume of patients than ADH and GH [21]. IALCH also has a higher bed capacity of 840 as compared to ADH and GH, further confirming that it services more patients between the three healthcare facilities [21]. Subsequently, it must be noted that in many instances, the patients are accompanied by family members when presenting to healthcare facilities [36]. Therefore, in addition to servicing a high volume of patients, OHPs deal with the added exposure to the emotional suffering and grief of the accompanying family members [2,3,8,14]. Another contributing factor is that IALCH does not only service the eThekwini metropolitan area, which is the largest between the two study settings in KZN, but also services patients that are referred from areas outside of eThekwini, which further increases the volume of patients serviced by IALCH [20,21]. The above is further supported by the findings of Mbeje et al., who reported that most (65.3%) of the new cancer cases (2307) in KZN that were reported in 2018 were from IALCH [37]. The study further reported that IALCH provided oncology services to the majority of the cancer patients between the three healthcare facilities [37]. Therefore, a plausible explanation would be that the higher the volume of patients, the higher the workload and occupational stress, which leads to a higher risk of CF; hence, the higher mean score for CF among OHPs from IALCH.

Commensurate with the proportion of females constituting the healthcare workforce, the number of female participants in this study was substantially higher than that of males, and this was reflected in the higher risk of CF among female participants, although the difference was not statistically significant [27,38]. Our results are consistent with those of other studies, which have also found that being female is associated with an increased risk of CF, as confirmed by Turgoose and Maddox in their narrative review on predictors of CF [39]. This may be attributable to women being generally more nurturing and caring than men and therefore likely exhibiting more compassion for others, as argued in the social-role theory of gender and helping [40,41]. The very nurturing and caring nature of women predispose them to increased risks of CF compared to their male counterparts, and the results of this study are congruent with that assertion.

This study found a significant positive correlation between the level of CF and age, suggesting that older OHPs are at an increased risk of CF. However, the positive correlation between CF and experience was not statistically significant. This finding is in contrast to the findings by Mason and Nel, which revealed that younger nurses were at a higher risk of CF [30]. They further suggested that younger caregivers are more susceptible to the effects of CF due to the lack of or the still developing skills and knowledge that are required to effectively cope with the nature of their work [30]. While one would expect longer work experience in the field to be associated with a lower risk of CF due to having developed effective coping abilities over the years, this study found a positive correlation between CF and longer work experience, although not statistically significant [39]. This study found that OHPs with more experience in oncology scored higher for CF. These findings are congruent with other similar studies conducted in the United States and China [10,42]. This could be attributable to other effects of working in the healthcare environment, as it is a demanding and high-stress environment that can bring about physical and mental exhaustion, which are associated with CF [10,43]. Thus, OHPs with more experience have been exposed to such occupational stressors and recurrent patient suffering for a longer period, which puts them at a higher risk of CF than their younger and less experienced counterparts, as was found in this study [3,8,44].

## 5. Limitations

This study was confined to OHPs from only the three public healthcare facilities, with a resultant small sample size. The use of total population sampling was intended to increase the sample size, although the final sample size was still relatively small, which is reflective of OHPs in the South African public health sector. However, the fact that not all OHPs participated in the study further affected the final sample size, and some questionnaires were not adequately completed, which resulted in their exclusion. Therefore, the generalisability of the findings of this study may be limited due to the small sample size. Another limitation is that, as this study was conducted during the COVID-19 pandemic, the impact this may have had on the results was not determined. Although this study provides adequate contextual data, additional research post pandemic with a larger sample size may provide broader and more generalisable data. The cross-sectional nature of this study means that the results only represent that single point in time, as the participants’ responses may be situational and, thus, could be different at another point in time. Therefore, longitudinal research in this regard may provide substantial contributions to the existing body of knowledge.

## 6. Conclusions

Despite this study coinciding with the COVID-19 pandemic, the prevalence of CF among OHPs was average compared to other studies. However, an average CF is a risk high enough to warrant action, as unmanaged CF may yield undesirable behavioural responses, including, but not limited to, absenteeism, something that can directly impact the healthcare system [45]. These behavioural responses may exacerbate pervasive staff shortages in healthcare facilities and ultimately affect patient care and the provision of adequate healthcare services; thus, interventions are necessary [45,46]. Intervention strategies dedicated to the management of CF among OHPs and healthcare professionals in general may be of benefit to not only the healthcare professionals, but also the patients that they care for. This study has highlighted the existence of the risk of CF among OHPs in the three public healthcare facilities that offer oncology services in KZN, South Africa. Evidence from this study is an important contribution to efforts pertaining to policy development and the planning of intervention strategies towards addressing CF among OHPs.

## Figures and Tables

**Figure 1 ijerph-20-05412-f001:**
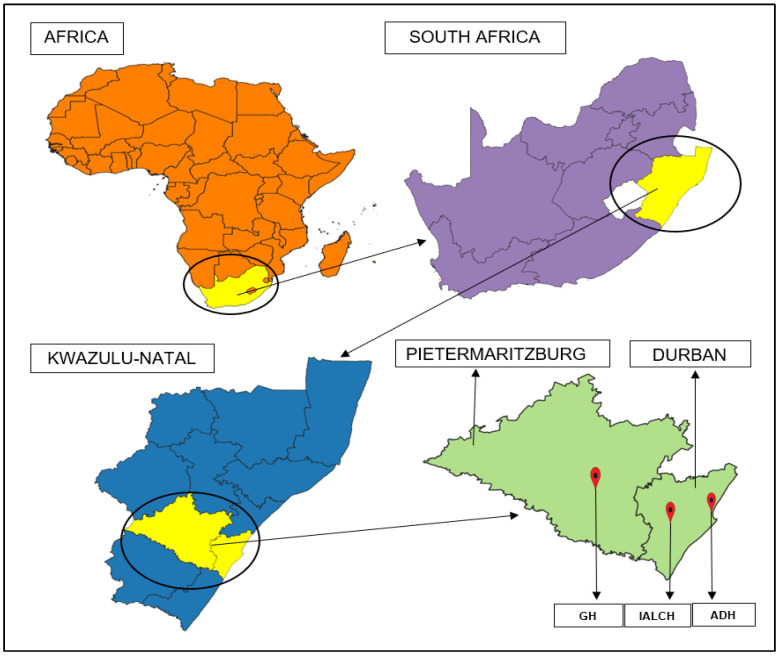
Location of the study sites within the broader regional context.

**Table 1 ijerph-20-05412-t001:** Demographic Characteristics of the Participants from the Three Public Healthcare Facilities.

Demographic Category		N	(%)
Age in years * Mean (SD) 39.16 (9.417)	20–30	13	17.8
	31–40	30	41.1
	41–50	14	19.2
	51–60	9	12.3
	Missing Values	7	9.6
	Total	73	100.0
Gender	Female	59	80.8
	Male	14	19.2
	Total	73	100.0
Occupation	Oncologist	8	10.9
	Nurse	34	46.6
	Radiographer	18	24.7
	Radiation Therapist	1	1.4
	Psychologist	2	2.7
	Medical Officer (Oncology)	1	1.4
	Other	9	12.3
	Total	73	100.0
Experience in years * Mean (SD) 11.45 (8.372)	1–5	15	20.5
	6–10	18	24.7
	11–20	25	34.2
	21–40	7	9.6
	Missing Values	8	11.0
	Total	73	100.0
Facility	AD	18	24.6
	IALCH	21	28.8
	GH	34	46.6
	Total	73	100.0

* Denotes missing values for category.

**Table 2 ijerph-20-05412-t002:** Overall Prevalence of Compassion Fatigue and Mean Scores by Healthcare Facility and Gender.

**Facility**	**CF Level**				
	**Low**	**Average**	**High**	**N**	**Mean Scores (SD)**
ADH	11	7	0	18	21.17 (5.752)
IALCH	4	17	0	21	26.76 (6.625)
GH	17	17	0	34	22.85 (6.933)
Total N (%)	32 (43.8)	41 (56.2)	0 (0.0)	73 (100)	23.56 (6.833)
*p* Value (F Value)					*p =* 0.025 (F = 3.879)
**Gender**	**Low**	**Average**	**High**	**N**	**Mean Scores (SD)**
Female	23	36	0	59	24.25 (6.506)
Male	9	5	0	14	20.64 (7.642)
*p* Value (T Value)					*p* = 0.075 (t = 1.805)

**Table 3 ijerph-20-05412-t003:** Pearson Correlation for CF Scores with Age and Experience.

Variable		Age	Experience	CF Score
Age	Pearson Correlation	-	0.802 **	0.279 *
	Sig. (2-tailed)		<0.001	0.023
Experience	Pearson Correlation		-	0.127
	Sig. (2-tailed)			0.313
CF score	Pearson Correlation			-
	Sig. (2-tailed)			

** Correlation is significant at the 0.01 level (2-tailed). * Correlation is significant at the 0.05 level (2-tailed).

## Data Availability

The data that support the findings of this study are available from the corresponding author, P.C.M., upon reasonable request.

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
