# Peer review of "The Prevalence of Compassion Fatigue among Oncology Healthcare Professionals in Three Public Healthcare Facilities in Kwazulu-Natal, South Africa"

_ijerph, 2023, doi:10.3390/ijerph20075412_

Round 1

Reviewer 1 Report

As strengths of the manuscript: a) the subject matter presented is relevant, useful and of transcendence in the health field. It is positively valued, b) the scope of application - three hospital centers - in a continent where the dissemination of their research is relatively scarce, contributes to the knowledge of the scientific community on this subject, and c) the inclusion in the study of health professionals with different specialties enriches it.

On the other hand, the variable selected for the study, compassion fatigue, is positively valued. However, the inclusion of other evaluation measures in the professionals, such as health-related quality of life, is lacking. That is, the scope of the research design is limited. It is understood that this handicap is insurmountable, based on the information provided in the Tools subsection.

With respect to the professionals, although the inclusion of different professionals is positively valued, a categorization of these professionals is lacking (as some categories have a very low number of participants), for example, according to the time of presence or type of relationship between professional and patient, given that the results could have enriched the study. This issue could be corrected in order to improve the design and quality of the work.

Another variable that, as a suggestion, could have been used is the experience in years of the professionals, studying the predictive character of the experience, in the sense of acting as a possible protective factor or, on the contrary, as a risk factor.

In conclusion, the manuscript is generally correct and useful, although the scope of the results is limited and the contribution of the same is scarce, for which reason it is recommended that the quality of the manuscript be improved as far as possible.

Reviewer 2 Report

The manuscript entitled:  The Prevalence of Compassion Fatigue Among Oncology Healthcare Professionals in Three Public Healthcare Facilities in Kwazulu-Natal, South Africa” reports a study principally aimed at exploring the prevalence of Compassion Fatigue among oncology healthcare professionals in three public healthcare facilities in South Africa, using the ProQOL-V Scale.

This is a phenomenon already found in previous studies. Therefore, the contribution of the paper to the literature is very limited. As a consequence, I encourage the authors to highlight the uniqueness of this study, the practical implications that the study results may have, perhaps even suggesting interventions for practitioners.

The article in its current form has some weaknesses. Here are some suggestions that I hope will help improve the article.

Firstly, with reference to the theoretical framework proposed by Stamm (2010), as the study is focused on one of the component of Compassion Fatigue, i.e. the Secondary Traumatic Stress, I suggest using the abbreviation CF/STS instead of just CF, as other literature also uses (e.g., Katsantoni, et al., 2019).

The introduction is repetitive and should be shortened by keeping the essential concepts.

Lines 62-64. Define the objectives of the study more broadly, adding the aims to analyse potential differences between health care facilities, gender etc.

Lines 100-107. Please, report approximately the number of staff members as in the case of the other two hospitals described. This is because the authors refer to the volume of patients and workload of the staff to comment on CF/STS differences among hospitals. Actually, the information on hospitals is very lengthy and lacks what could be really important information: data on oncology departments (rather than hospitals in general).

Lines 140-141. The language in which the scale was administered and more details about the Likert scale (from… to…) should be given. In addition, report the alpha indices for the subscales with reference to the sample of the present study.

Line 149 – The choice of cut offs is not clear. The sentence “The total cut-off scores for low, average and high CF were determined as follows:” should be replaced with: “The total cut-off scores suggested by the concise ProQOL 153 manual [23] for low, average and high CF were: …. These cut offs were used to determine the prevalence of CF/STS in the three hospitals”. As a consequence, the sentence on lines 153-154 should be removed.

Results

Because the CF/STS analyses were conducted on 73 participants who responded to this subscale of the ProQOL-V, the authors should describe the demographic characteristics of these 73 respondents (and not 84 study participants).

Authors should report all the values of the analyses used for comparisons (the t values for t-test analysis, the F values for ANOVAs), not just means and p values. Furthermore, post hoc would be expected with reference to ANOVAs.

Lines 218-229. Could it be that the professionals at IALCH are more "senior" than their colleagues at the other two hospitals? Have differences between hospitals been tested in terms of participant characteristics? If so, do any differences emerge? This is important to explain the differences of CF/STS.

Lines 241-243. The two sentences contradict each other: it is age, not oncology experience that correlates with CF/STS.

The fact that the data were collected during covid emergence is an issue to be addressed in the discussion (and not just mentioned in the study limits) because this may have heavily influenced OHPs' responses, so it should be considered in commenting on discrepancies with other data collected in previous studies.

Check Table 3 and substitute “,” with “.”

Round 2

Reviewer 1 Report

The theme of the study is very interesting and highly relevant to the scientific community. The authors have improved the quality of the manuscript, correcting the recommendations.

Reviewer 2 Report

Thank you for the accurate answers. The changes made to the manuscript have improved it.